The distribution and mitochondrial genotype of the hydroid Aglaophenia latecarinata is correlated with its pelagic Sargassum substrate type in the tropical and subtropical western Atlantic Ocean

Govindarajan Annette F. 1 afrese@whoi.edu
Cooney Laura 2
Whittaker Kerry 2
Bloch Dana 2
Burdorf Rachel M. 2
Canning Shalagh 2
Carter Caroline 2
Cellan Shannon M. 2
http://orcid.org/0000-0001-9378-5393 Eriksson Fredrik A.A. 2
Freyer Hannah 2
Huston Grayson 2
Hutchinson Sabrina 2
McKeegan Kathleen 2
Malpani Megha 2
Merkle-Raymond Alex 2
Ouellette Kendra 2
Petersen-Rockney Robin 2
Schultz Maggie 2
Siuda Amy N.S. 2 3
1 Woods Hole Oceanographic Institution , Woods Hole, MA , USA
2 Sea Education Association , Woods Hole, MA , USA
3 Marine Science Discipline, Eckerd College , St. Petersburg, FL , USA
Toonen Robert
Electronic publication date: 2019 Oct 18
Publication date: 2019
Volume: 7
Electronic Location ID: e7814
Received 2019 May 12; Accepted 2019 Sep 2
Copyright: © 2019 Govindarajan et al.
Copyright year: 2019
Copyright holder: Govindarajan et al.
License: This is an open access article distributed under the terms of the Creative Commons Attribution License, which permits unrestricted use, distribution, reproduction and adaptation in any medium and for any purpose provided that it is properly attributed. For attribution, the original author(s), title, publication source (PeerJ) and either DOI or URL of the article must be cited.
License URL: https://creativecommons.org/licenses/by/4.0/

Keywords: Hydrozoa, Sargassum, Hydroid, Sargasso Sea, Epiphytes, Aglaophenia

Funding: Sea Education Association, Eckerd College, the New England Aquarium Conservation Action Fund and the Virginia Wellington Cabot Foundation This research was funded by Sea Education Association, Eckerd College, the New England Aquarium Conservation Action Fund and the Virginia Wellington Cabot Foundation. The funders had no role in study design, data collection and analysis, decision to publish, or preparation of the manuscript.

==============================
The pelagic brown macroalga Sargassum supports rich biological communities in the tropical and subtropical Atlantic region, including a variety of epiphytic invertebrates that grow on the Sargassum itself. The thecate hydroid Aglaophenia latecarinata is commonly found growing on some, but not all, Sargassum forms. In this study, we examined the relationship between A. latecarinata and its pelagic Sargassum substrate across a broad geographic area over the course of 4 years (2015–2018). The distribution of the most common Sargassum forms that we observed (Sargassum fluitans III and S. natans VIII) was consistent with the existence of distinct source regions for each. We found that A. latecarinata hydroids were abundant on both S. natans VIII and S. fluitans III, and also noted a rare observation of A. latecarinata on S. natans I. For the hydroids on S. natans VIII and S. fluitans III, hydroid mitochondrial genotype was strongly correlated with the Sargassum substrate form. We found significant population genetic structure in the hydroids, which was also consistent with the distributional patterns of the Sargassum forms. These results suggest that hydroid settlement on the Sargassum occurs in type-specific Sargassum source regions. Hydroid species identification is challenging and cryptic speciation is common in the Aglaopheniidae. Therefore, to confirm our identification of A. latecarinata, we conducted a phylogenetic analysis that showed that while the genus Aglaophenia was not monophyletic, all A. latecarinata haplotypes associated with pelagic Sargassum belonged to the same clade and were likely the same species as previously published sequences from Florida, Central America, and one location in Brazil (São Sebastião). A nominal A. latecarinata sequence from a second Brazilian location (Alagoas) likely belongs to a different species.

Introduction

Sargassum, a common brown macroalgae, is distributed globally from temperate to tropical ocean waters. Of the more than 350 recognized species (Guiry & Guiry, 2018), Sargassum natans and S. fluitans are uniquely holopelagic (Butler et al., 1983; Stoner, 1983). These two species have several distinct forms that differ in their bladder and blade characteristics: S. natans is comprised of forms I, II, VIII, and XI, and S. fluitans is comprised of forms III and IV (Parr, 1939; Schell, Goodwin & Siuda, 2015). Sargassum natans and S. fluitans lack holdfasts and reproductive structures (Parr, 1939), and new individuals derive from the fragmentation of existing individuals adrift at the sea surface. Pelagic Sargassum is ecologically important as an oasis of life on the oligotrophic open ocean. Individual clumps, 10 s of centimeters in each dimension, host abundant epiflora and epifauna that serve as the base of a complex food web similar to those found in benthic habitats (Butler et al., 1983; Coston-Clements et al., 1991). Mats of aggregated clumps measuring 10 s of meters across additionally provide foraging or nursery habitat for fish (Wells & Rooker, 2004), turtles (Witherington, Hirama & Hardy, 2012), and seabirds (Moser & Lee, 2012).

Historically, pelagic Sargassum spp. was abundant in the Sargasso Sea and Gulf of Mexico and less abundant or absent in the Caribbean Sea (reviewed in Butler et al., 1983). Since 2011, coastlines on both sides of the tropical Atlantic, including the Caribbean Sea, have experienced three discrete and unprecedented inundations of pelagic Sargassum, each lasting many months (Schell, Goodwin & Siuda, 2015; Wang et al., 2019). The most recent event, during 2018, was the most extreme to date (Langin, 2018). Backtracking of landings using archived surface current model data (Franks, Johnson & Ko, 2016), satellite observations (Wang & Hu, 2017; Wang et al., 2019), and biophysical modeling (Brooks et al., 2018) indicate that these recent inundation events originated in the equatorial Atlantic, a new source region for pelagic Sargassum. In situ observations revealed that the inundating Sargassum was dominated by a previously rare form (S. natans VIII) that is morphologically distinct from the two common forms (S. natans I and S. fluitans III) of pelagic Sargassum observed in the Sargasso Sea (Schell, Goodwin & Siuda, 2015; Fig. 1).

Figure 1 Morphological differences between Sargassum forms as described in Parr (1939) and Schell, Goodwin & Siuda (2015).

S. fluitans have thorny stems whereas S. natans have smooth stems. Bladder and blade attributes differ widely among forms. (A) For S. fluitans III, thorns on stem are present, blades are wide, bladders are devoid of spines, and bladders are oblong. (B) For S. natans I, stem is smooth, blades are narrow, spines are present on bladders, and bladders are spherical (C) For S. natans VIII (photo credit: Janet Bering), stem is smooth, blades are wide, bladders are devoid of spines, and bladders are spherical.

Aglaophenia latecarinata (Allman, 1877) is a thecate hydroid that is commonly found on pelagic Sargassum (Calder, 1993; Calder, 1997; Cunha & Jacobucci, 2010; Fig. 2). In the Sargasso Sea, A. latecarinata is a dominant hydrozoan on S. fluitans III while it is rare or absent on S. natans I (Ryland, 1974; Niermann, 1986). Weis (1968) and Calder (1995) also report the absence of A. latecarinata on S. natans, though they do not report the type of S. natans they examined. A. latecarinata has been observed to be abundant, however, on S. natans VIII (Burkenroad in Parr, 1939). A. latecarinata is found on a variety of substrates in other parts of its range (Oliveira & Marques, 2007; Moura et al., 2018). Aglaophenia latecarinata forms feather-like colonies of polyps that can reach up to 10 mm in height (Calder, 1997). Within a colony, polyp fronds are connected via a stolon along the Sargassum substrate (Fig. 2). As the species lacks a medusa stage, the hydroids release planula larvae which settle onto nearby substrates and develop into new hydroids (Calder, 1997). For A. latecarinata on Sargassum substrates floating over deep ocean regions, planulae likely originate from the same or nearby Sargassum. Additionally, asexual fragments or propagules from the same or nearby Sargassum may also generate new hydroid colonies (Pati & Belmonte, 2018).

Figure 2 Aglaophenia latecarinata hydroids.

(A) Isolated Aglaophenia latecarinata specimen. (B) A colony of the epiphytic hydroid species, Aglaophenia latecarinata, attached to Sargassum stem. Arrow points to stolon connecting genetically identical units of a single colony.

Species identification based on morphology in aglaopheniid hydroids is exceptionally challenging (Svoboda & Cornelius, 1991). Recent genetic analyses suggest abundant cryptic speciation in the family and that A. latecarinata falls outside the main Aglaophenia clade (Postaire et al., 2016; Moura et al., 2018). The geographic range of A. latecarinata includes the northwestern Atlantic Ocean including the Sargasso Sea, the Gulf of Mexico, the southwestern Atlantic Ocean, and the western Pacific Ocean (Calder, 1997); however, A. latecarinata from the Sargasso Sea has not been included in any of the molecular analyses to date.

Here, we examined the relationship between A. latecarinata and its most common pelagic Sargassum substrates. We sought to determine whether we could observe population genetic variation as detected by the 16S gene over a vast geographic area in the tropical and subtropical western Atlantic Ocean, and if any observed variation could be associated with region and substrate type. Aglaophenia latecarinata was collected from both S. natans VIII and S. fluitans III from the tropical and subtropical western Atlantic between 2015 and 2018. We report the first observations of abundant A. latecarinata on S. natans VIII since the 1930s (Burkenroad in Parr, 1939) as well as a single observation of A. latecarinata on S. natans I. We sequenced the 16S gene from the hydroids on S. natans VIII and S. fluitans III and found that haplotypes were strongly associated with their Sargassum substrate type. We suggest that this finding could reflect hydroid colonization at different Sargassum source regions. We also show that in a family-level phylogenetic analysis of 16S sequences, A. latecarinata falls outside the main Aglaophenia clade and that wide-ranging pelagic Sargassum-associated A. latecarinata is the same species as individuals collected in previous studies from Florida, Central America, and Sao Sebastiao, Brazil.

Materials and Methods

Sampling

Aglaophenia latecarinata samples were collected aboard the SSV Corwith Cramer between 2015 and 2018 during Sea Education Association cruises from the Canary Islands to the Caribbean (2015), from San Juan, Puerto Rico to New York, New York (2015) or Woods Hole, Massachusetts (2016) and from Nassau, Bahamas to New York, New York (2017 and 2018) via Bermuda, or from San Juan, Puerto Rico to Key West, Florida (2018) and the cruise tracks were mapped using Ocean Data View 5.1.7 (Schlitzer, 2018) (Fig. 3). Cruise plans were filed with the US State Department, who obtained the required collection permits. No permits were required for sampling in international and US waters under federal jurisdiction. The cruise and permit numbers for the samples collected in this study are as follows: C-259 US State Department Cruise F2014-092, no permits necessary; C-263, US State Department Cruise F2015-044, no permits necessary; C-266, US State Department Cruise F2015-083, no permits necessary; C-273, US State Department Cruise F2016-084, Bermuda permit number SP170104, Bahamas MAMR/FIS/13; C-277, US State Department Cruise F2017-067, Haiti permit number SEMANAH/P-Nav/590, Dominican Republic permit (“Official Letter”) number 26940; and C279, US State Department Cruise F2017-112, Bermuda permit number SP171103 and Bahamas permit number MAMR/FIS/13.

Figure 3 Cruise tracks.

Each cruise is represented by a different color. Cruise labels are Sea Education Association cruise numbers. C259 and C263 took place in 2015, C266 took place in 2016, C273 took place in 2017, and C277 and C279 took place in 2018.

During each cruise, instruments mounted in line with a clean seawater flow-through system (intake at ~3 m) continuously measured temperature and salinity (Sea-Bird Electronics SBE 45 MicroTSG), as well as relative chlorophyll-a fluorescence (Turner Designs Model 10-AU in vivo chlorophyll-a fluorometer). Sargassum specimens were primarily collected opportunistically from dip nets that targeted distinct clumps. Less frequently, Sargassum clumps were collected from a neuston net (1 × 0.5 m) with 335 μm mesh towed alongside the ship at two knots for 30 min. We aimed to collect a maximum of 10–12 clumps of each of Sargassum form at every station. Each Sargassum clump was photographed and identified morphologically using Parr (1939) and Schell, Goodwin & Siuda (2015) (Fig. 1). When present, multiple A. latecarinata polyps from one individual hydroid (all connected by a visible stolon, Fig. 2) were plucked from each Sargassum clump using sterilized forceps, preserved in 95% ethanol as a single sample, and stored at room temperature. Vouchers of each Sargassum sample, including the representative hydroid and epibiont community, were preserved in 95% ethanol. Four representative vouchers that include A. latecarinata polyps attached to S. natans VIII and S. fluitans III substrates were submitted to the Smithsonian Museum of Natural History (USNM catalog numbers 1578893–1578896).

Sequencing and population analyses

For molecular analysis, we sequenced hydroids found on S. fluitans III and S. natans VIII. The hydroid colony on S. natans I was not included in our analysis given that it was a single observation. Two to three polyps from each hydroid individual were removed from ethanol, rinsed in deionized water, and diced using a sterilized razor blade. Genomic DNA (gDNA) of each hydroid sample was extracted using a Qiagen DNeasy Blood & Tissue Kit (Qiagen, Germantown, MD, USA) following the manufacturer’s protocol, and the final product was eluted twice using 100 µL of AE buffer. A segment of mitochondrial 16S rDNA was amplified using hydrozoan-specific primers: HYD1: 5′-TCG ACT GTT TAC CAA AAA CAT AGC-3′ and HYD2: 5′-ACG GAA TGA ACT CAA ATC ATG TAA G-3′ (Cunningham & Buss, 1993). PCR amplification consisted of an initial temperature of 95 °C for 3 min followed by 35 cycles at 95 °C for 30 s, 45 °C for 30 s and 68 °C for 60 s, and a final extension at 68 °C for 5 min. PCR products were visualized on a 1.5% agarose gel stained with SYBR® Safe (Invitrogen, USA). PCR products were purified using QIAquick PCR Purification kits (Qiagen, Germantown, MD, USA) following the manufacturer’s protocol except that the final elution step was modified to yield 30 µL total volume. Purified amplicons were quantified using a Nanodrop ND-1000 spectrophotometer (Thermo Fisher Scientific, Waltham, MA, USA) and sent to MWG Eurofins Operon (Huntsville, AL) or the DNA Analysis Facility at Yale University (New Haven, CT) for bidirectional Sanger sequencing on an ABI 3730XL capillary sequencer. Sequences were submitted to GenBank (accession numbers MK863834–MK863972).

Geneious versions 9.0.5 and 11.0.5 were used to assemble and curate chromatographs (Biomatters Ltd., Auckland, New Zealand). An alignment was generated with CLUSTALW (Larkin et al., 2007) using the default settings in Geneious. The alignment was trimmed at both ends to remove low-quality sequences. A haplotype network was constructed using TCS version 1.2.1 (Clement, Posada & Crandall, 2000) with gaps treated as a fifth character state.

Samples were categorized into broad oceanographic regions using QGIS (QGIS Development Team (2018) (Fig. 4). Samples collected south of 30°N (the approximate location of the subtropical convergence zone; Ullman, Cornillon & Shan, 2007) and north of the Greater Antilles were categorized as from the South Sargasso Sea. Samples collected north of 30°N and south of the Gulf Stream were categorized as from the North Sargasso Sea. Samples collected within the Gulf Stream were categorized as such. Samples collected south of 20°N in or out of the Caribbean were categorized as from the Caribbean or Tropical Atlantic, respectively. An analysis of molecular variance was performed and pairwise FST’s calculated with Arlequin ver. 3.5.2 (Excoffier & Lischer, 2010) to test for geographic structure between these regions.

Figure 4 Geographic distribution of hydroid samples by Sargassum form.

Lines indicate boundaries between oceanographic regions (Gulf Stream, North Sargasso Sea, South Sargasso Sea, Tropical Atlantic, and Caribbean) as described in the text. Each circle represents a single station and circle size corresponds to the number of Sargassum samples collected at each station. Yellow indicates stations where only S. fluitans III was collected, green indicates stations where only S. natans VIII was collected, and maroon indicates stations where both S. fluitans III and S. natans VIII were collected.

Phylogenetic analysis

To confirm A. latecarinata species identification, we downloaded representative 16S sequences for A. latecarinata and other aglaopheniid species from Genbank. Using ClustalW, we constructed an alignment with the Genbank sequences and one representative of each of our haplotypes from our short alignment. The ends of the alignment were trimmed and regions that contained gaps that could not be aligned with confidence were removed. Maximum likelihood and Bayesian analyses on this alignment were conducted using PAUP* (Swofford, 2003) and MrBayes (Huelsenbeck & Ronquist, 2001) through the Geneious interface. The best-fit model for these analyses was selected using Akaike Information Criterion using ModelTest (Posada & Crandall, 1998). In the maximum likelihood analysis, a phylogeny was constructed using the selected model and support for the nodes was determined with a bootstrap analysis with 1,000 replicates. The Bayesian analysis was run with a chain length of 1,100,000, a subsampling frequency of 1,000, a burn-in of 100,000, four heated chains, and a heated chain temperature of 0.2.

Results

Sampling

A total of 140 A. latecarinata colonies were collected at 47 stations across the Tropical Atlantic, Caribbean, South Sargasso Sea, North Sargasso Sea, and Gulf Stream regions (Fig. 4; Table S1) and sequenced. The distribution of A. latecarinata samples across two Sargassum forms was nearly even, with 68 colonies collected from S. natans VIII and 71 colonies collected from S. fluitans III. Only one A. latecarinata colony was observed on S. natans I (from the North Sargasso Sea in 2015) despite frequent observations of S. natans I throughout multiple cruises. The temporal and geographic distribution of Sargassum forms represented in the dataset was quite variable due to episodic S. natans VIII inundations from the equatorial Atlantic and predetermined cruise tracks (Table 1). However, because dip net sampling was selective, sample density is not necessarily representative of actual Sargassum density. For a representative view of Sargassum distribution and relative abundance, refer to Schell, Goodwin & Siuda (2015). In 2015, 20 of 21 A. latecarinata samples were collected from S. natans VIII at stations in the Tropical Atlantic. Of the 40 samples collected in 2016, a majority were again collected from S. natans VIII, and some from stations in the South Sargasso Sea (n = 20) and northern Gulf Stream (n = 12). In contrast, most (43 out of 46) samples in 2017 were collected from S. fluitans III at stations in the southern Gulf Stream (n = 5), South Sargasso Sea (n = 36) and North Sargasso Sea (n = 2). In 2018, 11 samples were collected from S. natans VIII and 14 samples were collected from S. fluitans III in the South Sargasso Sea, while six samples were collected from S. fluitans III in the North Sargasso Sea. The number of Sargassum clumps collected per station ranged from 1 to 12; only at four of 47 stations (C266-005, C266-011, C277-020, C279-003) were we able to concurrently collect A. latecarinata samples from S. natans VIII and S. fluitans III. These stations were all located in the South Sargasso Sea.

Table 1 Sample summary.

Number of hydroids samples collected from each Sargassum substrate by geographic region.

	2015	2016	2017	2018	Total	
	Sn8	Sf3	Sn8	Sf3	Sn8	Sf3	Sn8	Sf3	Sn8	Sf3	
South Sargasso	1	0	20	8	0	36	11	14	32	58	
North Sargasso	0	0	0	0	3	2	0	6	3	8	
Tropical Atlantic	20	0	0	0	0	0	0	0	20	0	
Gulf stream	0	0	12	0	0	5	0	0	12	5	
Caribbean	0	0	0	0	0	0	1	0	1	0	
Total	21	0	32	8	3	43	12	20	68	71	

Sequence variation

A 500 base pair alignment was obtained after trimming the sequence ends to remove low-quality regions. The alignment had eight variable positions, four of which were indels, that comprised 10 unique haplotypes. Haplotype frequency and diversity were strongly associated with algal substrate, with only one haplotype (haplotype 1) growing on both S. natans VIII and S. fluitans III (Fig. 5). Of the 68 A. latecarinata colonies found on S. natans VIII, 65 were haplotype 1, two were haplotype 2, and one was haplotype 3 (Fig. 5). Of the 71 A. latecarinata colonies found on S. fluitans III, two were haplotype 1, 44 were haplotype 4, one was haplotype 5, four were haplotype 6, one was haplotype 7, 13 were haplotype 8, one was haplotype 9, and five were haplotype 10 (Fig. 5). There were only two colonies collected from S. fluitans III that had an “S. natans VIII” haplotype (haplotype 1; these were 2016_SF3_41 from the South Sargasso Sea and 2017_SF3_65 from the Gulf Stream, Table S1). No colonies on S. natans VIII possessed an “S. fluitans III” haplotype. To confirm that our results were consistent over a longer sequence length, we constructed an alternative alignment with a subset of sequences that had expanded high-quality reads (590 base pairs for 99 sequences). The resulting haplotype analysis was consistent with the one based on the shorter, but more inclusive, alignment (Fig. S1).

Figure 5 Haplotype network of Aglaophenia latecarinata sequences.

Circle size reflects the number of individuals possessing a given haplotype (n). Yellow indicates hydroids found on S. fluitans III and green indicates hydroids found on S. natans VIII.

We found significant geographic structure between regions (Table 2). Pairwise FST values ranged from 0.02030 between the South and North Sargasso Seas, to 0.62631 between the North Sargasso Sea and the Tropical Atlantic (Table 3). All comparisons were significant (p < 0.05) except for between the South and North Sargasso Seas.

Table 2 Analysis of molecular variance (AMOVA) results.

Source of variation	df	Sum of squares	Variance components	Percentage of variation	
Among populations	3	6.171	0.07200 Va	19.77	
Within populations	135	39.433	0.29210 Vb	80.23	
Total	138	45.604	0.36410		
Fixation index	0.19774				

Table 3 Population pairwise FST’s (below the diagonal) and p-values (above the diagonal).

	TA	SS	NS	GS	
TA	–	0.00000 ± 0.0000	0.00000 ± 0.0000	0.01802 ± 0.0182	
SS	0.29003	–	0.22523 ± 0.0389	0.00901 ± 0.0091	
NS	0.62631	0.02030	–	0.00901 ± 0.0091	
GS	0.12084	0.10557	0.24759	–	
Note:

TA, Tropical Atlantic; SS, South Sargasso Sea; NS, North Sargasso Sea; GS, Gulf Stream.

Phylogenetic placement

We combined representative sequences of each of our A. latecarinata haplotypes with 16S sequences from aglaopheniid species on Genbank (Table 4). In instances when multiple sequences for a given species were present, we selected one sequence to represent that species, with the exception of A. latecarinata, for which we used all available sequences. The GenBank A. latecarinata sequences originated from two locations in the southern hemisphere (São Sebastião and Alagoas, Brazil), and five locations in the northern hemisphere ranging from Panama to Fort Pierce, Florida. The Florida sequence, which interestingly was the only specimen noted to originate from a hydroid colony on Sargassum, although the species of Sargassum was not given; (Moura et al., 2018), matched one of our haplotypes. The rest of the Genbank A. latecarinata sequences were unique. Our trimmed alignment was 378 base pairs. Our maximum likelihood and Bayesian phylogenetic analyses showed that the northern hemisphere sequences (including ours) formed a weakly supported clade (bootstrap and Bayesian posterior probability values were 66 and 0.69, respectively; Fig. 6). This clade, plus the São Sebastião sequence, formed a strongly supported clade (bootstrap and Bayesian posterior probability values were 98 and 1, respectively; Fig. 6). The A. latecarinata sequence from Alagoas, Brazil, fell outside of the main A. latecarinata clade and clustered strongly with A. rhynchocarpa (ML bootstrap and Bayesian posterior probability values were 100 and 1, respectively; Fig. 6). The genus Aglaophenia was not monophyletic although support for the arrangement of the nodes was generally weak—often less than 50 in the ML bootstrap analysis, but occasionally higher in the Bayesian analysis (Fig. 6).

Figure 6 Maximum likelihood phylogeny of the Aglaopheniidae.

Support for nodes is indicated by ML bootstrap values before the slash (only those above 50 are shown) and Bayesian posterior probabilities after the slash (only those above 0.5 are shown). A. latecarinata sequences are highlighted in red.

Table 4 Aglaopheniid sequences from GenBank used in the phylogenetic analysis.

Genus	Species	Reference	GenBank accession#	Location (A. latecarinata)	
Aglaophenia	latecarinata	Moura et al. (2018)	MH212420	Carrie Bow Key, Belize	
Moura et al. (2018)	MH212421	Ft. Pierce, FL, USA
(on Sargassum)	
Moura et al. (2018)	MH212422	Isla Tambor, Panama	
Moura et al. (2018)	MH212423	Isla Uvita, Costa Rica	
Moura et al. (2018)	MH212424	Key Biscayne, FL, USA	
Maronna et al. (2016)	KT266600	Barra do São Miguel, Alagoas, Brazil	
Leclère, Schuchert & Manuel (2007)	DQ855936	Ponta do Baleeiro, São Sebastião, Brazil	
acacia	Leclère et al. (2009)	FJ550507		
cupressina	Postaire et al. (2016)	KM587399		
elongata	Leclère et al. (2009)	FJ550508		
harpago	Moura et al. (2012)	JN560129		
kirchenpaueri	Moura et al. (2012)	JN560124		
lophocarpa	Moura et al. (2012)	JN560112		
octodonta	Leclère, Schuchert & Manuel (2007)	DQ855915		
parvula	Moura et al. (2012)	JN560097		
picardi	Moura et al. (2012)	JN560105		
pluma	Moura et al. (2012)	JN560130		
postdentata	Postaire et al. (2016)	KM587408		
rhynchocarpa	Maronna et al. (2016)	KT266601		
sinuousa	Postaire et al. (2016)	KM587411		
struthionides	Maronna et al. (2016)	KT266602		
tubiformis	Leclère, Schuchert & Manuel (2007)	DQ855917		
tubulifera	Moura et al. (2012)	JN560117		
sp. 1 (CM 2011)	Moura et al. (2012)	JN560094		
sp. 2 (CM 2011)	Moura et al. (2012)	JN560101		
Cladocarpus	cartieri	Moura et al. (2012)	JN560085		
integer	Leclère et al. (2009)	FJ550512		
Gymnangium	allmani	Postaire et al. (2016)	KM587415		
eximium	Postaire et al. (2016)	KM587417		
gracilicaule	Postaire et al. (2016)	KM587438		
hians	Postaire et al. (2016)	KM587446		
montagui	Moura et al. (2012)	JN560075		
sp. 1 (BP 2015)	Postaire et al. (2016)	KM587460		
Lytocarpia	brevirostris	Boissin et al. (2018)	MH108512		
canepa	Maronna et al. (2016)	KT266645		
myriophyllum	Moura et al. (2012)	JN560089		
nigra	Postaire et al. (2016)	KM587482		
phyteuma	Postaire et al. (2016)	KM587489		
sp. 1 (BP 2015)	Postaire et al. (2016)	KM587509		
Macrorhynchia	philippina	Postaire et al. (2016)	KM587516		
phoenicea	Postaire et al. (2016)	KM587526		
sibogae	Postaire et al. (2016)	KM587537		
spectabilis	Postaire et al. (2016)	KM587539		
sp. 1 (BP 2015)	Postaire et al. (2016)	KM587538		
sp. 2 (BP 2015)	Postaire et al. (2016)	KM587510		
Streptocaulus	multiseptatus	Moura et al. (2012)	JN560080		
dollfusi	Moura et al. (2012)	JN560081		
Aglaopheniidae	sp. (CM 2011)	Moura et al. (2012)	JN560079		
Note:

Sampling locations are given for A. latecarinata sequences.

Discussion

Our results add to growing evidence from field and satellite observations that Sargassum forms have different source regions and dispersal patterns (Gower & King, 2011; Gower, Young & King, 2013; Schell, Goodwin & Siuda, 2015; Wang & Hu, 2017). The distinct but overlapping geographic ranges of S. fluitans III and S. natans VIII observed in Schell, Goodwin & Siuda (2015) are also evident in our pattern of opportunistic sample collection. Sargassum natans VIII was abundant in the Tropical Atlantic and also present in the northern Gulf Stream during some years. Sargassum natans VIII and S. fluitans III co-occurred in the South Sargasso Sea.

Prior to 2011, the primary forms of Sargassum in the Sargasso Sea were S. fluitans III and S. natans I (Schell, Goodwin & Siuda, 2015). Satellite data from this time suggests these forms range from the Caribbean to the Sargasso Sea and likely originate from the Gulf of Mexico (Gower & King, 2011). In contrast, S. natans VIII is likely transported from its source in the North Equatorial Recirculation Region via the North Equatorial Current, which splits into the Caribbean Current in the Caribbean Sea and the Antilles Current that runs north of the Greater Antilles (Franks, Johnson & Ko, 2016; Putman et al., 2018; Brooks et al., 2018; Wang et al., 2019). Sargassum natans VIII from the Caribbean may then be carried into the Gulf of Mexico and on to the Gulf Stream, where it could mix and travel north along with the S. natans VIII from the Antilles Current (Wang et al., 2019). Because the boundary between the Antilles Current and South Sargasso Sea is weak, both S. fluitans III and S. natans VIII are commonly found in this region (Schell, Goodwin & Siuda, 2015).

The seasonality of the blooms and their associated dispersal may also contribute to the maintenance of distinct distributions of each Sargassum type and their associated A. latecarinata genotypes. Both the Gulf of Mexico and the tropical Atlantic exhibit Sargassum blooms in spring (Wang et al., 2019). We suggest that the Gulf of Mexico Sargassum (which likely includes S. fluitans III) may be exported to the North Atlantic before the tropical Atlantic Sargassum (which is likely S. natans VIII) arrives there, thus minimizing the potential for interaction of their associated hydroids. Additional sampling of both the hydroids and the Sargassum in the Gulf of Mexico and equatorial Atlantic source regions will be necessary to test our proposed mechanism for maintaining distinct A. latecarinata populations.

Our frequent observations of A. latecarinata on S. fluitans III and S. natans VIII and single observation of A. latecarinata on S. natans I are consistent with previous findings. Burkenroad (in Parr, 1939) reported that A. latecarinata (reported as A. minuta) was the dominant hydroid on both S. fluitans III and S. natans VIII. Ryland (1974) and Niermann (1986) also did not report A. latecarinata on their surveys of epibionts on S. natans I. Weis (1968) did not find A. latecarinata on S. natans and Calder (1995) identified A. latecarinata as the dominant hydrozoan on S. fluitans, but noted that it was entirely absent from S. natans. Neither Weis (1968) nor Calder (1995) specified the type of S. natans that they observed, but we suggest that it was likely the S. natans I form. Settlement specificity of hydroid species has been observed both within and between other Sargassum species (Nishihira, 1965; Nishihira, 1971; Calder, 1995).

The species-specific substrate pattern that we observed could be due to several factors including substrate selection by planula larvae or substrate availability. For example, larvae of the epiphytic hydroid Coryne uchidae (Stechow) showed larval settlement preferences when presented with multiple algal substrates including different non-pelagic Sargassum species (Nishihira, 1968a). Substrate selection in C. uchidae appears to be influenced by chemical cues present in substrate extracts (Nishihira, 1968b; Kato et al., 1975). We are not aware of any similar experiment with A. latecarinata larvae. However, A. latecarinata hydroids have been successfully transplanted on to S. natans I (Burkenroad in Parr, 1939) suggesting that the hydroids can grow on this Sargassum form if settlement occurs. If the different Sargassum species have distinct source regions as recent research suggests (Schell, Goodwin & Siuda, 2015; Franks, Johnson & Ko, 2016; Wang & Hu, 2017), and larval settlement occurs in these source regions where the Sargassum species are not found together, it is possible that substrate availability is primarily responsible for our, and previous, observations. Hydroid colonies could also potentially originate asexually via dislodged fragments attaching to Sargassum, and this could lead to species-specific patterns if re-attachment were substrate-specific. More research is needed to determine the degree of substrate specificity as well as the mechanisms driving any specificity, both for pelagic A. latecarinata and for aglaopheniids in general.

For the S. fluitans III and S. natans VIII, which had abundant A. latecarinata colonization, we found a striking correlation between hydroid mitochondrial genotype and Sargassum species. We also found significant population genetic structure in A. latecarinata between North Atlantic regions, which was likely due to the distribution of the Sargassum substrates. These findings are consistent with the hypothesis that substrate availability associated with Sargassum species source regions is driving A. latecarinata colonization patterns. However, the detection of two S. fluitans III-derived A. latecarinata colonies in haplotype 1, which was the most common haplotype for S. natans VIII-derived colonies, points to the potential for limited genetic exchange to take place when the Sargassum forms co-occur. We found that S. fluitans III and S. natans VIII were sometimes found together at the same sampling site in a given year, as was the case for one of the S. fluitans III-derived A. latecarinata colonies with haplotype 1. While we did not simultaneously collect S. natans VIII along with the other S. fluitans III-derived A. latecarinata colony with haplotype 1, it is possible that it had encountered S. natans VIII previously as it drifted through other regions. As such, the dominant population genetic pattern observed in A. latecarinata is likely maintained through settlement on substrates that differ in geographic origins.

Aglaophenia species typically lack a planktonic medusa stage (Svoboda & Cornelius, 1991) and have been hypothesized to have limited dispersal capabilities, which could lead to population genetic structuring and speciation (Postaire et al., 2016). However, benthic or fixed stages can also disperse via rafting on algal or other substrates (Ronowicz, Kukliński & Mapstone, 2015; Boissin et al., 2018). In contrast to the pattern observed in A. latecarinata, the Sargassum shrimp Latreutes fucorum, which has a long-lived planktonic larval period, exhibited no population genetic structure over the same region (Sehein et al., 2014). This finding could reflect the greater potential of L. fucorum to disperse independently of Sargassum, and reinforces our hypothesis that substrate availability is an important driver of A. latecarinata settlement patterns.

Several studies that utilize 16S sequences suggest cryptic species are common in aglaopheniid taxa. Schuchert (2014) found that in the aglaopheniid genus Plumularia, nominal morphologically-defined species showed a high degree of genetic variability indicating possible cryptic speciation. Postaire et al. (2016), employing multiple species delimitation methods, found significant variation in mitochondrial 16S sequences within aglaopheniid morphospecies, which also likely indicates cryptic species. In another study of the family, Moura et al. (2012) found that their 16S A. latecarinata sequence from Brazil fell outside of the Aglaophenia clade. Our family-level genetic analysis suggests that the different pelagic Sargassum-associated hydroid genotypes represent intraspecific variation and not cryptic species. Furthermore, our results show that these Sargassum-associated specimens are also likely the same species as those collected from Central America and Florida in Moura et al. (2018) and from São Sebastião in Brazil (Leclère, Schuchert & Manuel, 2007). However, the sequence from Alagaos, Brazil (Maronna et al., 2016) fell well outside the A. latecarinata clade and so may represent a cryptic species or a misidentification. At a deeper level, our phylogeny indicated that the genus Aglaophenia is polyphyletic, as in previous studies (Moura et al., 2018). Taxonomic studies, coupled with molecular analyses utilizing multiple genetic markers, will be necessary to fully understand aglaopheniid diversity and evolutionary relationships.

Conclusions

Aglaophenia latecarinata hydroids were abundant on S. natans VIII and S. fluitans III, but rare on S. natans I. For the hydroids on S. natans VIII and S. fluitans III, hydroid mitochondrial genotype was strongly correlated with Sargassum substrate form. There was significant population genetic structure in the hydroids, which likely reflects the distribution of their different algal substrates, with S. natans VIII likely annually sourced primarily from the equatorial Atlantic and S. fluitans III likely annually sourced primarily from the Gulf of Mexico. As cryptic speciation appears to be common in aglaopheniids, we conducted a family-level phylogenetic analysis that showed that the genus Aglaophenia was polyphyletic, and that all A. latecarinata haplotypes associated with pelagic Sargassum belonged to the same clade as published sequences from Florida, Central America, and one location in Brazil (São Sebastião). A nominal A. latecarinata sequence from a second Brazilian location (Alagoas) likely belongs to a different species.

Supplemental Information

Supplemental Information 1 Haplotype network based on a longer alignment (590 base pairs) and fewer specimens (99 sequences).

Circle size reflects the number of individuals possessing a given haplotype (n). Yellow indicates hydroids found on S. fluitans III and green indicates hydroids found on S. natans VIII.

Click here for additional data file.

Supplemental Information 2 List of specimens and associated data.

Click here for additional data file.

We thank students and crew of SEA Semester (Sea Education Association, Woods Hole, MA, USA) cruises C259, C263, C266, C273, C277, and C279 for help with Sargassum collection.

Additional Information and Declarations

Competing Interests

Author Contributions

Field Study Permissions

Data Availability

The authors declare that they have no competing interests.

Annette F. Govindarajan conceived and designed the experiments, performed the experiments, analyzed the data, contributed reagents/materials/analysis tools, prepared figures, and/or tables, authored or reviewed drafts of the paper, approved the final draft.

Laura Cooney conceived and designed the experiments, performed the experiments, analyzed the data, contributed reagents/materials/analysis tools, prepared figures, and/or tables, authored or reviewed drafts of the paper, approved the final draft.

Kerry Whittaker conceived and designed the experiments, performed the experiments, analyzed the data, contributed reagents/materials/analysis tools, prepared figures, and/or tables, authored or reviewed drafts of the paper, approved the final draft.

Dana Bloch performed the experiments, analyzed the data, approved the final draft.

Rachel M. Burdorf performed the experiments, analyzed the data, approved the final draft.

Shalagh Canning performed the experiments, analyzed the data, approved the final draft.

Caroline Carter performed the experiments, analyzed the data, approved the final draft.

Shannon M. Cellan performed the experiments, analyzed the data, approved the final draft.

Fredrik A.A. Eriksson performed the experiments, analyzed the data, approved the final draft.

Hannah Freyer performed the experiments, analyzed the data, approved the final draft.

Grayson Huston performed the experiments, analyzed the data, approved the final draft.

Sabrina Hutchinson performed the experiments, analyzed the data, approved the final draft.

Kathleen McKeegan performed the experiments, analyzed the data, approved the final draft.

Megha Malpani performed the experiments, analyzed the data, approved the final draft.

Alex Merkle-Raymond performed the experiments, analyzed the data, approved the final draft.

Kendra Ouellette performed the experiments, analyzed the data, approved the final draft.

Robin Petersen-Rockney performed the experiments, analyzed the data, approved the final draft.

Maggie Schultz performed the experiments, analyzed the data, approved the final draft.

Amy N.S. Siuda conceived and designed the experiments, performed the experiments, analyzed the data, contributed reagents/materials/analysis tools, prepared figures, and/or tables, authored or reviewed drafts of the paper, approved the final draft.

The following information was supplied relating to field study approvals (i.e., approving body and any reference numbers):

Cruise plans were filed with the US State Department, who obtained the required collection permits. No permits were required for sampling in international and US waters under federal jurisdiction.The cruise and permit numbers for the samples collected in this study are as follows: C-259 US State Department Cruise F2014-092, no permits necessary; C-263, US State Department Cruise F2015-044, no permits necessary; C-266, US State Department Cruise F2015-083, no permits necessary; C-273, US State Department Cruise F2016-084, Bermuda permit number SP170104, Bahamas MAMR/FIS/13; C-277, US State Department Cruise F2017-067, Haiti permit number SEMANAH/P-Nav/590, Dominican Republic permit (Official Letter) number 26940; and C279, US State Department Cruise F2017-112, Bermuda permit number SP171103 and Bahamas permit number MAMR/FIS/13.

The following information was supplied regarding data availability:

Sequences are available at GenBank: MK863834–MK863972.

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
