# Peer review of "The distribution and mitochondrial genotype of the hydroid Aglaophenia latecarinata is correlated with its pelagic Sargassum substrate type in the tropical and subtropical western Atlantic Ocean"

_PeerJ, doi:10.7717/peerj.7814_

## Round 0.1 · original submission · Major Revisions

I now have comments back from 2 expert referees, both of which are encouraging about your study but also raise concerns about your manuscript. The first referee is quite enthusiastic but also points out that you actually have little data to explicitly test your main conclusion about sources. I tend to agree and feel that you either need to address this issue to the satisfaction of the referees or ease up on the conclusion about source populations in the absence of actual data. The second referee questions the use of 16S as the marker of choice for this study. Again this seems reasonable to me given the literature to date and worthy of additional discussion or validation to convince future readers that the results and conclusions are robust, Given each of the referees have substantial concerns about either the data or major conclusions of the study, I am inclined to return a decision of major revisions to ensure that these concerns are adequately addressed in the revision of the manuscript before it is likely to become acceptable for publication.

Reviewer 1 ·

Basic reporting

The language is clear and to the point.
The literature references are complete.
The new sequence data have been assigned Genbank accession numbers.
The voucher specimens, with DNA aliquots, have not been deposited in a museum where they could be accessed by future researchers. Nor is it indicated how these vouchers could be obtained. This situation should be rectified.
The article structure is clear and of professional standard.
The MS is a good publishable unit.

Experimental design

The work is original and entirely worth publishing. It will make a nice contribution from PeerJ. The MS provides evidence that population structure of a hydroid species that colonizes different Sargassum species/forms is likely explained by colonization of the different species/forms in different source regions, with only rare transfer from one Sargassum type to another. The picture that emerges fits in with limited prior information on larval settlement of hydroids. The results are relatively straight forward. The MS includes a quick phylogenetic analysis of mt16S sequences for various species of Aglaopheniidae. They identify a likely misidentification of one specimen/sequence from Brazil. The authors are careful to indicate that it might have been cryptic, but the zero length branch it forms with A. rhynchocarpa suggests a misidentification. For the phylogenetic analysis, the choice to remove all "regions that contained gaps"is probably more conservative than needed. Nonetheless, it is not likely to impact any of the results in a meaningful way. Aglaophenia will still be polyphyletic, but including more data might bring out the population level structuring a bit more in the hypothesized topology.

Validity of the findings

The MS is a well written description of an interesting piece of research. They conclude from the data they present that the settlement of one particular hydroid species associated with Sargassum likely occurs in type – specific Sargassum source regions, and that this can explain observed population structuring of the hydroid species.

Additional comments

It may seem obvious, but it could be nice if the authors were more explicit about how to specifically test their main conclusion. What do we know about the source regions of the Sargassum species/forms? Can one go to them and find and sample A. latecarinata? If these two species of Sargassum are holopelagic, how does A. latecarinata colonize them? From the benthos?

Reviewer 2 ·

Basic reporting

See general comments to authors

Experimental design

See general comments to authors

Validity of the findings

See general comments to authors

Additional comments

The distribution and mitochondrial genotype of the hydroid Aglaophenia latecarinata is correlated with
its Sargassum substrate type in the Sargasso Sea


In this paper the authors produced ~140 sequences of the hydroid Aglaophenia latecarinata from two different forms of Sargassum. They use one single gene and produce what they call “a family level tree” using additional sequences from genbank.
My first concern with the paper is that the 16S gene is not suitable for family level phylogenies, and in fact has rarely (if never) been used with that intent. It is a good barcoding molecule (similar to COI for most organisms), and it does a pretty good job in identifying species boundaries, but it is not very useful for higher-level phylogenies. It is not very good for population studies as well. The discussion about the fact that A. latecarinata falls outside the genus Aglaophenia, or the monophyly of the genus, is of little value, in my opinion (just look at the bootstrap values and at the posterior probabilities of most of the clades in the tree).
Other claims in the paper are not well substantiated. Line 251 to 254 authors say that their data confirm previously published work that shows A. latecarinata as the dominant hydroid on Sargassum. It is my understanding that the authors did not survey other hydroids, they just collected Aglaophenia. Thus it is not clear how they came to the conclusion that A. latecarinata is the dominant Hydrozoa.
So in summary, what the paper is showing is that using one single gene (which is not ideal for population analysis) there are soma haplotypes that are mainly found on one Sagassum type, and some that are found on the other. The finding is somewhat interesting, but limited in scope. I will let the editor decide whether this finding is sufficient to grant publication in PeerJ.
Other comments:
In the introduction it would be good to have an explicit question that the authors are trying to answer with this paper.
The discussion on the species-specific substrate pattern (starting at Line 261) does not take in consideration asexual reproduction (asexual pieces of the colony can attach themselves to the substrate). It should probably be mentioned in this context.
It would be useful to know (if possible) the substrate of other A. latecarinata in the phylogenetic tree (that were obtained from genbank) to expand the discussion a little.
Species names in the phylogenetic tree should be in italic
In abstract, A. latecarinata is misspelled (A. latecarina)

---

## Round 0.2 · accepted · Accept

I have now read your revised manuscript and feel that you have addressed the concerns of the referees sufficiently that there is no need for additional review. There are a few final revisions that will need to be completed prior to publication (such as you USNM catalog numbers) but these are minor enough that they can be added during production. Thanks for submitting your work to PeerJ!